# AN EFFICIENT ALGORITHM FOR COMPUTING OPTIMAL WASSERSTEIN BALL CENTER

## ABSTRACT

Wasserstein Barycenter (WB) is a fundamental problem in machine learning, whose objective is to find a representative probability measure that minimizes the sum of Wasserstein distance to given distributions. WB has a number of applications in various areas. However, in some applications like model ensembling, where it aggregates predictions of different models on the label space, WB may lead to unfair outcome towards underrepresented groups (e.g., a "minority" distribution may be far away from the obtained WB under Wasserstein distance). To address this issue, we propose an alternative objective called "Wasserstein Ball Center (WBC)". Specifically, WBC is a distribution that encompasses all input distributions within the minimum Wasserstein distance, which can be formulated as a minmax optimization problem. We show that the WBC problem with fixed support is equivalent to solving a large-scale linear programming (LP) instance, which is quite different from the previously studied LP model for WB. By incorporating some novel observations on the induced normal equation, we propose an efficient algorithm that accelerates the interior point method by $O(Nm)$ times ($N$ is the number of distributions and $m$ is the support size). Finally, we conduct a set of experiments on both synthetic and real-world datasets. We demonstrate the computational efficiency of our algorithm, and showcase its better accuracy on model ensembling under imbalanced data distributions.

## 1 INTRODUCTION

To find a representative of several given probability distributions is a natural problem in machine learning. One popular approach is to compute the geometric center on probability space with induced distances between probabilities, such as the Wasserstein distance ((Villani, 2021)). Given a weight vector $(\omega_1, \omega_2, \ldots, \omega_N)$ for $N \geq 2$, the **Wasserstein barycenter** (WB) of $N$ probability measures $\{\mu_k\}_{k=1}^N$ is defined as the weighted Frechet mean under Wasserstein distance. Namely, it is the solution of the following problem

$$\min_{\mu \in \mathcal{P}_p(\Omega)} \sum_{k=1}^{N} \omega_k \mathcal{W}_p^p(\mu, \mu_k), \qquad (1)$$

where $\mathcal{P}_p(\Omega)$ is the set of Borel probability measure on $\Omega$ with finite p-th moment, and $\mathcal{W}_p$ is the Wasserstein distance of order $p$, which will be formally defined in Section 2. WB has found various applications in many fields, such as economics (Carlier & Ekeland, 2010; Chiappori, 2017), physics (Benamou et al., 2014; Koehl et al., 2019), statistics (Goldfeld et al., 2024; Backhoff-Veraguas et al., 2022; Kroshnin et al., 2021), and machine learning (Dognin et al., 2019; Zhuang et al., 2022; Cheng et al., 2021).

As the Frechet mean under Wasserstein distance, WB tends to assign more measure to the region where the input density functions "cluster". In other words, to minimize the average distance from the barycenter to the input probabilities, if the support of most distribution is concentrated with high probability in a region, then the WB should also have measure concentrated in that region. But this property may behave "unfairly" to "minority", *i.e.* the distributions with support deviated from the majority of others could be too far away from the WB. Figure 1 gives an intuitive demonstration for this issue.

The unfairness could cause negative impact in some scenarios. To shed some light, we take the application of WB in model ensembling as an example (Dognin et al., 2019; Lin et al., 2023; Qin et al., 2021). The high-level idea of model ensembling is as follows. In multi-class prediction, our task is to train a model that outputs a probability vector where each coordinate corresponds to a semantic class. If we obtain multiple such models, then WB can be adopted as an appropriate candidate to ensemble them, because it usually exhibits better generalization than simple arithmetic and geometric mean, due to its diversity and smoothness (Dognin et al., 2019). However, the prediction models can be trained separately with quite different datasets (Wen et al., 2020). If there is an "outlier" dataset distinguished from others, the model trained on it could be neglected in this WB-induced ensemble model.

To address this unfair issue, we propose a different objective function. Rather than minimizing the summation of Wasserstein distances, we try to find a distribution that is of minimal distance from the farthest input distribution:

$$\min_{\mu \in \mathcal{P}_p(\Omega)} \max_{k \in [N]} W_p(\mu, \mu_k). \tag{2}$$

From a geometric perspective, we can think of it as the center of the ball in Wasserstein space, who covers all input distributions with minimum radius. In this setting, the output distribution does not put extra measure to the region where input distributions cluster with high density. Please see Figure 1 for an illustrative comparison. We call the solution for Problem (2) the **Wasserstein Ball Center (WBC)**, and aim to design an efficient algorithm for solving it. It should be noted that "Wasserstein ball" is not a new concept and actually has been studied by several works before (Yue et al., 2022; Pesenti & Jaimungal, 2023; Chen et al., 2024), yet these previous works usually assume the ball center is given and take the ball as a feasible region for constraining some optimization objective. But in this paper, we focus on how to compute an optimal center so that the induced radius (under Wasserstein distance) is minimized.

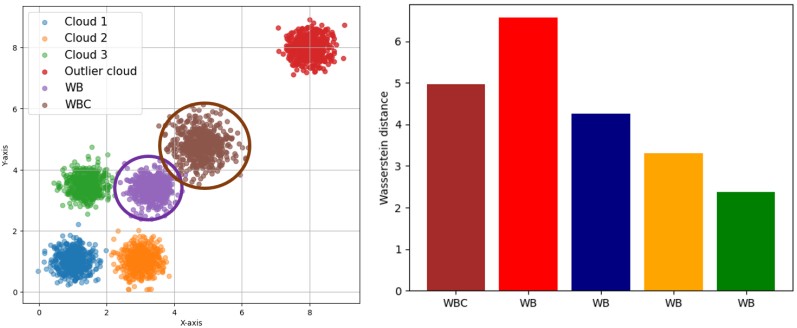

Figure 1: Four probability measures, with their WB enclosed in purple ellipse, WBC enclosed in brown ellipse. Note that the red cloud has measures distributed distinctly from the others. In the histogram on the right, the y-axis denote the Wasserstein distance to the WBC. We show that WBC treats the outlier more equally, while keeps the other three clustered distributions adequately near.

### 1.1 OUR MAIN CONTRIBUTIONS

Solving the problem WBC (2) is not an easy job due to its "minmax" nature, more specifically, it is challenging to find a proper subgradient for its objective function. When all distributions are of discrete support, the problem can be formulated as a linear programming (LP) problem, where the details are shown in Section 2. Partly inspired by the recent interior point method (IPM) based algorithms for solving the WB problem (1) (e.g., (Ge et al., 2019)), we also consider to develop an efficient IPM based algorithm for the WBC problem, though the formulation for WB has a much simpler structure without the minimax issue.

Technically, there are several significant challenges for directly applying IPM to the WBC problem, *e.g.,* the computational cost and space complexity are both very large. The linear programming formulation of WBC has $m \sum_{i=1}^{N} m_i + m + N + 1$ variables and $Nm + \sum_{i=1}^{N} m_i + N + 1$ constraints, where the integer $N$ denotes number of distributions, $m_i$ and $m$ denote the size of support for the

$i$-th distribution and WBC respectively. This brings the challenge that to compute the inner loop of IPM requires a time complexity of $O(N^3(m_i + m)^3)$. To tackle this difficulty, we grind the intrinsic information of constraint matrix to simplify the Newton normal equation, which is a linear system with a large positive definite constraint matrix, and is the most expensive part in each inner loop of IPM. Specifically, we simplify the matrix inverse occurred in the solution of Newton path, based on an important observation:

*The seemingly dense matrix can be decomposed into a sum of two matrices, one is block diagonal, and the other is a matrix that is highly duplicated, implying low rank.*

Then, we can apply the renowned Woodbury's equality (Hager, 1989) to reduce the complexity for inversion of the sum of a simple matrix and a low-rank matrix. We obtain a $\mathbf{O(N^2 m^3)}$ time complexity for each iteration, whereas the vanilla IPM requires $\mathbf{O(N^3 m^4)}$ by straight matrix inversion (for simplicity we just assume $m_i = O(m)$ here). The latter one is beyond acceptable scope in many real-world scenarios. For example, for a problem that $N = 10^2$ and the order of magnitude of $m = 10^3$. The complexity of our algorithm is $10^{13}$, while the vanilla IPM requires $10^{18}$, which is $10^5$ times higher. The formal description on this result is presented in Theorem 3.2. We also conduct a set of experiments to evaluate our algorithm. As for the practical effectiveness, our algorithm can be significantly faster than the popular commercial solver Gurobi. For example, if given an instance with $N = 1000, m = 100$, our implementation can solve the problem in $5$ minutes while Gurobi takes about $18$ minutes, on a workstation with Intel(R) Core(TM) i5-9400 CPU.

### 1.2 RELATED WORKS

**Wasserstein distance.** The Wasserstein distance, also known as the Earth Mover's distance when $p = 2$, quantifies the dissimilarity between two probability distributions, particularly when their supports are discrete sets. Computing the discrete Wasserstein distance actually is equivalent to solving a min-cost max flow problem (Ahuja et al., 1991; Khesin et al., 2021). Several more efficient discrete Wasserstein distance algorithms were proposed, such as (Ling & Okada, 2007; Pele & Werman, 2009). It is also a classic topic in machine learning (Rüschendorf, 1985; Pele & Werman, 2009). By using matrix scaling technique, Cuturi (2013) introduced the "Sinkhorn Distance", which incorporates an entropic regularization term to smooth the transportation problem, offering significantly faster solutions than exact computation of the discrete Wasserstein distance. Following Cuturi's work, recent years have seen the development of several improved Sinkhorn algorithms (Lin et al., 2019; Altschuler et al., 2019; Benamou et al., 2015; Altschuler et al., 2017).

**Wassertein barycenter.** Cuturi & Doucet (2014) showed that the computation for WB can be improved by using an entropic regularization, leading to a simple gradient-descent scheme that was later improved and generalized under the iterative Bregman projection (IBP) algorithm (Benamou et al., 2015). Further progress includes the semi-dual gradient descent (Cuturi & Peyré, 2018), accelerated primal-dual gradient descent (APDAGD) (Kroshnin et al., 2019), alternating direction method of multipliers (ADMM) (Ye et al., 2017), deterministic IBP (Lin et al., 2020), and the IPM algorithm MAAIPM (Ge et al., 2019).

**Interior Point Method.** The interior point method was discovered by Dikin (1967). The method was reinvented in 1984, when Karmarkar developed a method for linear programming called "Karmarkar's algorithm" that runs in polynomial time (Karmarkar, 1984). Since then IPM has attracted a great amount of attention, where one of the most successful IPM methods is the class of primal-dual approaches. Mehrotra's predictor-corrector algorithm (Mehrotra, 1992) provides the basis for most implementations of this class of methods, which is also the type of IPM applied in this paper (the details of predictor-corrector IPM are presented in Section 3.2). (Mizuno et al., 1993) proposed the Mizuno-Todd-Ye method, which has the best iteration complexity $O(\sqrt{nL})$ and quadratic convergence (Ye et al., 1993). For more information on IPM, we refer the reader to the survey paper (Gondzio, 2012). Recently, there are also some new studies on reducing the exponent of IPM in theoretical computer science (Jiang et al., 2020; Cohen et al., 2021), which relies on a technique called "matrix maintenance" to reduce the update time for each iteration.

**Fairness and class imbalance.** The fairness issue has attracted a great amount of attention in machine learning (Joseph et al., 2016; Mehrabi et al., 2021; Caton & Haas, 2024). The proposed solutions include adjusting labels from sensitive groups to reconstruct unbiased mapping (Dwork et al., 2012; Jiang & Nachum, 2020), and removing sensitive attributions (Krasanakis et al., 2018). Our work was inspired by *socially fair clustering* (Ghadiri et al., 2021; Makarychev & Vakilian, 2021),

which proposed an objective to minimize the maximal distances from the centers to groups. It is also connected with class imbalance of data. Unfairness can result from the issue of representation bias, which arises due to insufficient amount of data in certain groups or subgroups (Lohaus et al., 2020; Chai & Wang, 2022). Existing methods include fair data generation (Jang et al., 2021), multi-objective optimization (Martinez et al., 2020) and boosting (Gong & Kim, 2017).

## 2 PRELIMINARIES

For two discrete probability vectors $\boldsymbol{u} \in \mathbb{R}_{n_1}, \boldsymbol{v} \in \mathbb{R}_{n_2}$, define the set of matrices $\mathcal{M}(\boldsymbol{u}, \boldsymbol{v}) = \{\Pi \in \mathbb{R}_{+}^{n_1 \times n_2} : \Pi \mathbf{1}_{n_2} = \boldsymbol{u}, \Pi^{\top} \mathbf{1}_{n_1} = \boldsymbol{v}\}$ as the coupling matrices, which consists of all joint distributions of margin $\boldsymbol{u}$ and $\boldsymbol{v}$. Let $\mathcal{Q} = \{(a_i, \boldsymbol{q}_i) : i = 1, \ldots, m\}$ denote the discrete probability measure supported on $m$ points $\boldsymbol{q}_1, \ldots, \boldsymbol{q}_m$ in $\mathbb{R}^d$ with weights $a_1, \ldots, a_m$ respectively. The **Wasserstein distance** of the two discrete probability measures $\mathcal{Q} = \{(a_i, \boldsymbol{q}_i) : i = 1, \ldots, m_1\}$ and $\mathcal{P} = \{(b_j, \boldsymbol{p}_j) : j = 1, \ldots, m_2\}$ is

$$\mathcal{W}_p(\mathcal{Q}, \mathcal{P}) := \min \left\{ (\sum_{i=1}^{m_1} \sum_{j=1}^{m_2} \pi_{ij} \|\boldsymbol{q}_i - \boldsymbol{p}_j\|_p^p)^{\frac{1}{p}} : \Pi = [\pi_{ij}] \in \mathcal{M}(\boldsymbol{a}, \boldsymbol{b}) \right\} \quad (3)$$

where $\boldsymbol{a} = (a_1, \ldots, a_{m_1})^{\top}$ and $\boldsymbol{b} = (b_1, \ldots, b_{m_2})^{\top}$. A set of probability measure $\{\mathcal{P}^{(t)}, t = 1, \cdots, N\}$ is denoted by $\mathcal{P}^{(t)} = \{(a_i^{(t)}, \boldsymbol{q}_i^{(t)}) : i = 1, \ldots, m_t\}$, with probability vector $\boldsymbol{a}^{(t)} = (a_1^{(t)}, \ldots, a_{m_t}^{(t)})^{\top}$. The optimal **Wasserstein ball center** (WBC) $\mathcal{P}_{opt} = \{(w_i, \boldsymbol{x}_i) : i = 1, \cdots, m\}$ is another probability measure such that the maximum Wasserstein distance to these given $N$ probability measures is minimized, as defined in the objective function (2) when $\Omega = \{\boldsymbol{x}_1, \cdots, \boldsymbol{x}_m\}$.

The probability $w$ of $\mathcal{P}_{opt}$ and its coupling matrices with $\{\boldsymbol{a}^{(t)} : t = 1, \cdots, N\}$ must be in a solution set $\mathcal{S} = \{(\boldsymbol{w}, \Pi^{(1)}, \ldots, \Pi^{(N)}) \in \mathbb{R}_{+}^m \times \mathbb{R}_{+}^{m \times m_1} \times \cdots \times \mathbb{R}_{+}^{m \times m_N} : \mathbf{1}_m^{\top} \boldsymbol{w} = 1, \boldsymbol{w} \geq 0; \Pi^{(t)} \mathbf{1}_{m_t} = \boldsymbol{w}, (\Pi^{(t)})^{\top} \mathbf{1}_m = \boldsymbol{a}^{(t)}, \Pi^{(t)} \geq 0, \forall t = 1, \cdots, N\}$. For a given support $\Omega$, the distance matrices is defined as $\mathcal{D}^{(t)}(\Omega) = (\|\boldsymbol{x}_i - \boldsymbol{q}_j^{(t)}\|_p^p)_{(i,j)} \in \mathbb{R}^{m \times m_t}$ for $t = 1, \ldots, N$. Then Problem (2) is equivalent to

$$\min_{\boldsymbol{w}, \Omega, \Pi^{(t)}} \max_{t \in [N]} \left\langle \mathcal{D}^{(t)}(\Omega), \Pi^{(t)} \right\rangle \quad \text{s.t.} \quad (\boldsymbol{w}, \Pi^{(1)}, \ldots, \Pi^{(N)}) \in \mathcal{S}, \ \boldsymbol{x}_1, \ldots, \boldsymbol{x}_m \in \mathbb{R}^n. \quad (4)$$

For most practical applications, we can assume that all measures in $\{\mathcal{P}^{(t)}\}_{t=1}^N$ have the same set of support points, and the barycenter should also take the same set of support points (e.g., fixed support WB (Dognin et al., 2019)). Thus we focus on the case when the support $\Omega$ is given. This fixed-support assumption turns WBC into the following linear programming:

$$\min_{\boldsymbol{w}, \Pi^{(t)}} \max_{t \in [N]} \left\langle \mathcal{D}^{(t)}, \Pi^{(t)} \right\rangle \quad \text{s.t.} \quad (\boldsymbol{w}, \Pi^{(1)}, \ldots, \Pi^{(N)}) \in \mathcal{S} \quad (5)$$

where $\mathcal{D}^{(t)}$ denotes $\mathcal{D}^t(\Omega)$ for simplicity. To make the LP formulation clear, we use slack variable $\gamma \in \mathbb{R}$, turning problem (5) into the following

$$\min_{\boldsymbol{w}, \Pi^{(t)}, \gamma} \gamma$$
$$\text{s.t.} \ (\boldsymbol{w}, \Pi^{(1)}, \ldots, \Pi^{(N)}) \in \mathcal{S}, \ \boldsymbol{x}_1, \ldots, \boldsymbol{x}_m \in \mathbb{R}^n \quad (6)$$
$$\left\langle \mathcal{D}^{(t)}(X), \Pi^{(t)} \right\rangle \leq \gamma, \ 1 \leq t \leq N.$$

## 3 OPTIMIZATION FRAMEWORK FOR WBC

In this section, we introduce our optimization framework for WBC. Specifically, we first formalize WBC to be the standard LP form and ensure that the constraint matrix is full row-rank in Section 3.1. In Section 3.2, we introduce the IPM framework we implement. In Section 3.3, we illustrate how to eliminate unnecessary computations in IPM.

## 3.1 PRECONDITIONING

We use $\texttt{vec}(A)$ to denote the vectorization of a matrix $A$. To reduce the problem to the standard-form linear program, we vectorize the constraints $\Pi^{(t)}\mathbf{1}_{m_t} = \boldsymbol{w}$ and $\left(\Pi^{(t)}\right)^\top \mathbf{1}_m = \boldsymbol{a}^{(t)}$ to be:

$$(\mathbf{1}_{m_t}^\top \otimes I_m)\texttt{vec}(\Pi^{(t)}) = \boldsymbol{w}, \quad (I_{m_t} \otimes \mathbf{1}_m^\top)\texttt{vec}(\Pi^{(t)}) = \boldsymbol{a}^{(t)}, \quad t = 1, \cdots, N.$$

Thus, Problem (6) is formulated as:

$$\min \ \boldsymbol{c}^\top \boldsymbol{x} \quad \text{s.t.} \ A\boldsymbol{x} = \boldsymbol{b}, \boldsymbol{x} \geq 0 \tag{7}$$

with $\boldsymbol{x} = (\texttt{vec}(\Pi^{(1)}); ...; \texttt{vec}(\Pi^{(N)}); \boldsymbol{w}; \gamma_1; \ldots; \gamma_N, \gamma)$, $\gamma_i = \gamma - \langle D^{(i)}, \Pi^{(i)}\rangle$, $b = (\boldsymbol{a}^{(1)}; ...$
$\boldsymbol{a}^{(N)}; \mathbf{0}_m; ...; \mathbf{0}_m; 1)$, $\boldsymbol{c} = (0, \ldots, 1)$ and $A = \begin{bmatrix} E_1 \\ E_2 \ E_3 \\ \mathbf{1}_m^\top \\ D \quad I_N \ -\mathbf{1}_N \end{bmatrix}$, where $E_1 = \texttt{diag}(I_{m_1} \otimes$
$\mathbf{1}_m^\top, ..., I_{m_N} \otimes \mathbf{1}_m^\top)$, $E_2 = \texttt{diag}(\mathbf{1}_{m_1}^\top \otimes I_m, ..., \mathbf{1}_{m_N}^\top \otimes I_m)$, $E_3 = -\mathbf{1}_N \otimes I_m$ and $D = \texttt{diag}(\texttt{vec}(\mathcal{D}^{(1)}), \ldots, \texttt{vec}(\mathcal{D}^{(N)}))$. Let $M := \sum_{i=1}^N m_i$, and then we have $n_c := Nm + M + N + 1$ constraints and $n_v := mM + m + N + 1$ variables. Based on these notations, we know that the problem can be written as a standard form LP with $n_v$ variables and $n_c$ constraints.

To implement IPM, it is essential that $A$ is of full row-rank. We defer the reason to the next section. The following lemma eliminates all redundant constraints, turning $A$ into a full row-rank matrix $\bar{A}$. Specifically, $\bar{A} \in \mathbb{R}^{(n_c-N)\times n_v}$ is the matrix obtained from $A$ by removing the $(M+1)$-th, $(M+m+1)$-th, $\cdots$, $(M+(N-1)m+1)$-th rows of $A$, and $\bar{\boldsymbol{b}} \in \mathbb{R}^{n_c-N}$ be the vector obtained from $\boldsymbol{b}$ by removing the $(M+1)$-th, $(M+m+1)$-th, $\cdots$, $(M+(N-1)m+1)$-th entries of $\boldsymbol{b}$.

**Lemma 3.1.** *1)* $\bar{A}$ *has full row-rank; 2) solving the equation* $A\boldsymbol{x} = \boldsymbol{b}$ *is equivalent to solving the equation* $\bar{A}\boldsymbol{x} = \bar{\boldsymbol{b}}$.

Due to Lemma 3.1 (proof of which is left in appendix), we can now focus on $\bar{A}$ instead of $A$ in the following subsections.

## 3.2 PREDICTOR-CORRECTOR IPM

We choose the classic predictor-corrector scheme (Mehrotra, 1992; Wright, 1997), which was also applied previously to accelerate the computation for WB (Ge et al., 2019). As a second order method, it is proved to have quadratic convergence rate (Ye et al., 1993), which surpasses first-order methods. When we deal with a primal-dual system of linear programming, from Karush–Kuhn–Tucker theory, we have search direction found by applying a Newton-like method to equations. The equations are in the following system with current *barrier parameter* $\mu_+$, which is taken as the coefficient of a logarithm barrier function. Writing in matrix form, the search direction at a feasible point $(\boldsymbol{x}, \boldsymbol{y}, \boldsymbol{s})$ should be the solution of the following nonlinear system of equations:

$$\begin{bmatrix} 0 & \bar{A}^\top & I \\ \bar{A} & 0 & 0 \\ S & 0 & X \end{bmatrix} \begin{bmatrix} \Delta\boldsymbol{x} \\ \Delta\boldsymbol{y} \\ \Delta\boldsymbol{s} \end{bmatrix} = \begin{bmatrix} \bar{A}^\top \boldsymbol{y} + \boldsymbol{s} - \boldsymbol{c} \\ \bar{A}\boldsymbol{x} - \boldsymbol{b} \\ -X\boldsymbol{s} + \Delta X \Delta \boldsymbol{s} + \mu_+ \mathbf{1} \end{bmatrix}, \tag{8}$$

where $\boldsymbol{y}, \boldsymbol{s}$ are dual variables for the constraints $A\boldsymbol{x} = \boldsymbol{b}$ and $\boldsymbol{x} \geq 0$ respectively, and $X$ is a diagonal matrix with $X_{ii} = \boldsymbol{x}_i$. To reduce this nonlinear system to linear cases, first we obtain a *predictor step* by removing the "$\Delta X\Delta\boldsymbol{s} + \mu_+ \mathbf{1}$" term on the RHS of eq. (8), then compute the *corrector step* by assigning the predictor steps to the RHS of eq. (8). For a fixed $\mu_+$, we update the solution by step descent with corrector step until convergence, then update $\mu_+$ to a smaller value and do above procedure all over again. As $\mu_+$ approaches to 0, the current position convergences to the optimal solution of LP (7).

The solution of Eq. (8) can be obtained by sequentially computing $\Delta\boldsymbol{y}, \Delta\boldsymbol{s}$ and $\Delta\boldsymbol{x}$, where their values are detailed in Appendix B. In both predictor and corrector steps, the hardest part is to compute $\Delta\boldsymbol{y}$, as the solution of

$$(ARA^\top)\Delta\boldsymbol{y} = \boldsymbol{f}, \tag{9}$$

where $R = \texttt{diag}(\boldsymbol{s})^{-1}X$, $\boldsymbol{f}$ is a vector computed at last step. Equation (9) is often referred as **normal equation** (Wright, 1997), and we elaborate on our idea for solving it in the next section.

## 3.3 SOLVING THE NORMAL EQUATIONS EFFICIENTLY

In this section, we introduce an efficient algorithm for solving the normal equation $(\bar{A}R\bar{A}^\top)\Delta\boldsymbol{y} = \boldsymbol{f}$, whose complexity is summarized in the following theorem.

**Theorem 3.2.** *There exists an IPM algorithm, such that in each inner iteration, the time complexity in terms of flops is $O(m^2 \sum_{i=1}^{N} m_i + Nm^3 + N^2m^2 + N^3)$.,*

**Roadmap of the proof.** To prove Theorem 3.2, we need to simplify the reverse of $\bar{A}R\bar{A}^\top$. Proposition 3.3 illustrates the structure of $\bar{A}R\bar{A}^\top$, lemma 3.4 essentially reduces the ranks of blocks of $\bar{A}R\bar{A}^\top$. Lemma C.1 and lemma 3.5 analyse how to break some matrix inverses into simple forms, turning a multiplication between one vector with a big matrix into that with multiple small matrices, and give respective time complexity.

Let $\boldsymbol{r}$ be the $n_v$-dimensional vector with its $i$-th entry $r_i = R_{ii}$. Let $M_2 = N(m-1)$, which is the rank of the matrix $(E_2\ E_3)$. First, we present the basic block-wise structure of $\bar{A}R\bar{A}^\top$.

**Proposition 3.3.** *Let $\boldsymbol{z} = \boldsymbol{r}(Mm+1 : Mm+m)$. $\bar{A}R\bar{A}^T$ can be written as the following format:*

$$\bar{A}R\bar{A}^T = \begin{bmatrix} B_1 & B_2 & \mathbf{0} & K_1 \\ B_2^\top & B_3+B_4 & \boldsymbol{\alpha} & K_2 \\ \mathbf{0} & \boldsymbol{\alpha}^\top & c & 0 \\ K_1^\top & K_2^\top & 0 & W \end{bmatrix}$$

*where $B_1 \in \mathbb{R}^{M \times M}$ is a diagonal matrix with positive diagonal entries; $B_2 \in \mathbb{R}^{M \times M_2}$ is a block-diagonal matrix with $N$ blocks, the $i$-th block is of size $m_i \times (m-1)$; $B_3 \in \mathbb{R}^{M_2 \times M_2}$ is a diagonal matrix with positive diagonal entries, then $B_4 = (\mathbf{1}_N\mathbf{1}_N^\top) \otimes \mathtt{diag}(\boldsymbol{z})$; $\boldsymbol{\alpha} = -\mathbf{1}_N \otimes \boldsymbol{z}$; $c = \mathbf{1}_m^\top \boldsymbol{r}(n_v - m + 1 : n_v - N)$. $K_1 \in \mathbb{R}^{M \times N}$ is a block-diagonal matrix with $N$ blocks, with the $i$-th block of size $m_i \times 1$, $K_2 \in \mathbb{R}^{M_2 \times N}$ is a block-diagonal matrix with $N$ blocks, with the $i$-th block of size $(m-1) \times 1$. $W = W_1 + r_{n_v}\mathbf{1}\mathbf{1}^\top$, $W_1 \in \mathbb{R}^{N \times N}$ is a diagonal matrix with positive diagonal entries.*

*Proof.* All through direct computation. The identity $(U_1 \otimes V_1)(U_2 \otimes V_2) = (U_1U_2) \otimes (V_1V_2)$ (when the RHS exists) can be used to simplify the computation. $\square$

Now we simplify the coefficient matrix $\bar{A}R\bar{A}^\top$ of the linear system by performing several elementary transformation, such that it turns into a block diagonal matrix. Then we solve the system with the transformed coefficient matrix, and finally transform the obtained solution back for the original solution of $(\bar{A}R\bar{A}^\top)\boldsymbol{z} = \boldsymbol{f}$. Define

$$Q_1 := \begin{bmatrix} I_M & & & \\ -B_2^\top B_1^{-1} & I_{M_2} & & \\ & & 1 & \\ & & & I_N \end{bmatrix}, \quad Q_2 := \begin{bmatrix} I_M & & & \\ & I_{M_2} & -\boldsymbol{\alpha}/c & \\ & & 1 & \\ & & & I_N \end{bmatrix}, \quad Q_3 := \begin{bmatrix} I_M & & & \\ & I_{M_2} & & \\ & & 1 & \\ -B_1^{-1}K_1^\top & & & I_N \end{bmatrix}.$$

Let $A_1 := B_3 - B_2^\top B_1^{-1} B_2$ and $A_2 := B_4 - \frac{1}{c}\boldsymbol{\alpha}\boldsymbol{\alpha}^\top$, $\bar{K}_2$ and $\bar{W}$ in $Q_3$ are the matrices in place of $K_2$ and $W$ after eliminating $B_2$ and $K_1$ by applying $Q_1$ and $Q_2$ to $\bar{A}R\bar{A}^\top$. Then, we have the transformation:

$$Q_3Q_2Q_1\bar{A}R\bar{A}^T Q_1^\top Q_2^\top Q_3^\top =: \begin{bmatrix} B_1 & & \\ & A_1+A_2 & c & \bar{K}_2 \\ & \bar{K}_2^\top & \bar{W} \end{bmatrix}.$$

Now we want to eliminate $\bar{K}_2$ in order to obtain a block diagonal matrix that is easy to invert. Therefore, we need to compute $Q_4 := \begin{bmatrix} I_M & & & \\ & I_{M_2} & & \\ & & 1 & \\ & -\bar{K}_2^\top(A_1+A_2)^{-1} & & I_N \end{bmatrix}$. With some calculation, we have the following lemma.

**Lemma 3.4.**

*1. $A_2 = (\mathbf{1}_N\mathbf{1}_N^\top) \otimes Z$, where $Z = \mathtt{diag}(\boldsymbol{Z}) - \frac{1}{c}\boldsymbol{z}\boldsymbol{z}^\top$ ($\boldsymbol{z}$ is the vector defined in proposition 3.3),*

*2. $A_1$ is a block-diagonal matrix with $N$ blocks $A_{ii}$. The size of each block is $(m-1) \times (m-1)$.*

According to lemma 3.4(1), let $A_1 = \mathtt{diag}(A_{11}, A_{22}, ..., A_{NN})$, where each $A_{ii} \in \mathbb{R}^{(m-1) \times (m-1)}$.

**Lemma 3.5.**

$$(A_1 + A_2)^{-1} = A_1^{-1} - A_1^{-1}\left((\mathbf{1_N 1_N^\top}) \otimes (Z^{-1} + \sum_{i=1}^{N} A_{ii}^{-1})^{-1}\right)A_1^{-1}. \tag{10}$$

*The time complexity for applying a vector to the RHS of eq.* (10) *is* $O(Nm^2)$.

*Proof.* We defer the proof of eq (10) to appendix. For the time complexity, notice that (1) $A_1^{-1}$ is a diagonal matrix with only $N(m-1)^2$ nonzero term, thus multiplying a vector to it costs no more than $O(Nm^2)$. (2)$(\mathbf{1_N 1_N^\top} \otimes (Z^{-1} + \sum_{i=1}^{N} A_{ii}^{-1})^{-1})$ duplicates $N^2$ copies of $(Z^{-1} + \sum_{i=1}^{N} A_{ii}^{-1})^{-1}$. Therefore, if you multiply by a column vector on the right, it will result in $N$ set of identical operations. The same applies to left multiplication. □

To eliminate $\bar{K}_2$, $\bar{W}$ will be replaced by $\tilde{W} = \bar{W} - \bar{K}_2^\top (A_1 + A_2)^{-1}\bar{K}_2$. This is done in only $O(N^2m^2)$ time, since $\bar{K}$ can be viewed as $N$ vectors.

Now we are ready to present Algorithm 1. The following algorithm is a step-by-step procedure for solving the normal equation given the above diagonalized coefficient matrix.

---

**Algorithm 1:** Solver for $(\bar{A}R\bar{A}^\top)\Delta\mathbf{y} = \mathbf{f}$

---

**Input:** $R \in \mathbb{R}^{n_v \times n_v}$, $\mathbf{f} \in \mathbb{R}^{n_c - N}$ as described in eq. (9).
**Output:** The solution $\Delta\mathbf{y}$.

1 Compute $B_1, B_2, B_3, K_1, K_2, W$;            // Initialization
2 Compute $Q_1, Q_2, Q_3$;
3 $\bar{K}_2 \leftarrow K_2 - B_2^\top B_1^{-1}K_1$, $\bar{W} \leftarrow W - K_1^\top B_1^{-1}K_1$; ;       // Eliminate $K_1$
4 Compute $Q_3, A_1, A_2$;
5 $\mathbf{z}^{(1)} \leftarrow Q_1\mathbf{f}, \mathbf{z}^{(2)} \leftarrow Q_2\mathbf{z}^{(1)}, \mathbf{z}^{(3)} \leftarrow Q_3\mathbf{z}^{(2)}$; // Process RHS of eq. (9) in sync
6 Decompose $(A_1 + A_2)^{-1}$ according to Lemma 3.5 ;
7 Compute $Q_4, \mathbf{z^{(4)}} \leftarrow Q_4\mathbf{z}^3$;
8 $\tilde{W} \leftarrow \bar{W} - \bar{K}_2^\top (A_1 + A_2)^{-1}\bar{K}_2$; ;          // Eliminate $\bar{K}_2$
9 $\mathbf{z}^{(5)}(1:M) \leftarrow B_1^{-1}\mathbf{z}^{(4)}(1:M)$ ;        // First M rows of $\mathbf{z}^{(4)}$
10 $\mathbf{z}^{(5)}(n_c - N + 1 : n_c) \leftarrow \tilde{W}^{-1}\mathbf{z}^{(4)}(n_c - N + 1 : n_c)$; $\mathbf{z}^{(5)}(n_c - N) \leftarrow c^{-1}\mathbf{z}^{(4)}(n_c - N)$;
   // Last N+1 rows of $\mathbf{z}^{(5)}$
11 Compute $(A_1 + A_2)\mathbf{z}^{(4)}(M + 1 : n_c - N - 1) = z^{(3)}(M + 1 : n_c - N - 1)$
;                          // other entries of $\mathbf{z}^{(5)}$
12 $\mathbf{z}^{(6)} \leftarrow Q_4^\top \mathbf{z}^{(5)}$, $\mathbf{z}^{(7)} \leftarrow Q_3^\top\mathbf{z}^{(6)}$, $\mathbf{z}^{(8)} \leftarrow Q_2^\top\mathbf{z}^{(7)}$, $\Delta\mathbf{y} \leftarrow Q_2^\top\mathbf{z}^{(8)}$ ;     // recover $\Delta\mathbf{y}$.
13 **return** $\Delta\mathbf{y}$

---

With Lemma 3.5 tackling the hardest parts of algorithm 1, theorem 3.2 can be easily concluded.

*Proof.* We count the flops required in each step in Algorithm 1:

$$step\,1 : O(m\sum_{t=1}^{N} m_t)); \; step\,2 : O(1); \; step\,3 : O(N^2m^2); \; step\,4 : O(\sum_{t=1}^{N} m_t)$$

$$step\,5 : O(Nm + \sum_{t=1}^{N} m_t); \; step\,6 : O(Nm^3 + Nm\sum_{t=1}^{N} m_t); \; step\,7 : O(N^m); \; step\,8 : O(N^2m^2)$$

$$step\,9 : O(M); \; step\,10 : O(N^3); \; step\,11 : O(Nm^3); \; step\,12 : O(Nm\sum_{t=1}^{N} m_t)$$

The computation of step 3, step 6 and step 10 requires most flops. □

## 4 EXPERIMENTS

### 4.1 COMPUTATIONAL EFFICIENCY

We conduct three experiments to investigate the real performance of our algorithm. **(1)** The first experiment demonstrates our advantages on computational speed and memory usage over commercial solver Gurobi, a powerful optimization solver widely used across various fields such as operations research, finance, and data science. **(2)** The second experiment reflects the fairness of WBC over the standard WB. For these two experiments, the entries of the weight of $(q_1^{(t)}, ..., q_m^{(t)})$ in distribution $\mathcal{P}^{(t)}$ are generated uniformly at random. **(3)** The third experiment further illustrates the performance on a real-world dataset FairFace (Karkkainen & Joo, 2021) with considering the racial issue. We choose 700 (100 for each race) images including seven racial groups of "Black", "East Asian", "Indian", "Latino-Hispanic", "Middle Eastern", "Southeast Asian" and "White" as 700 distributions (each image actually can regarded as a distribution). All the experiments are implemented on a workstation, Intel(R) Core(TM) i5-9400 CPU @ 2.90GHz and 8GB for RAM, equipped with win64 - Windows 11+.0.

The baseline we choose is *Gurobi Optimizer version 11.0.0* (academic license) . **Comparison with Gurobi** Firstly, we conduct two experiments to compare the computational performance of our method and Gurobi, then conduct another two experiments to show the computational performance when the variables size $Nm^3$ grows over $10^5$. Without loss of generality, we set $m$ of all distributions to be equal for brevity.

As Fig. 2 shows, our algorithm is always faster than Gurobi, and the gap between the two methods is expanding as the scale increases. Moreover, Gurobi can not solve the instance with $m > 500$ due to memory limitation, which showcases the superiority of the space complexity in Theorem 3.2.

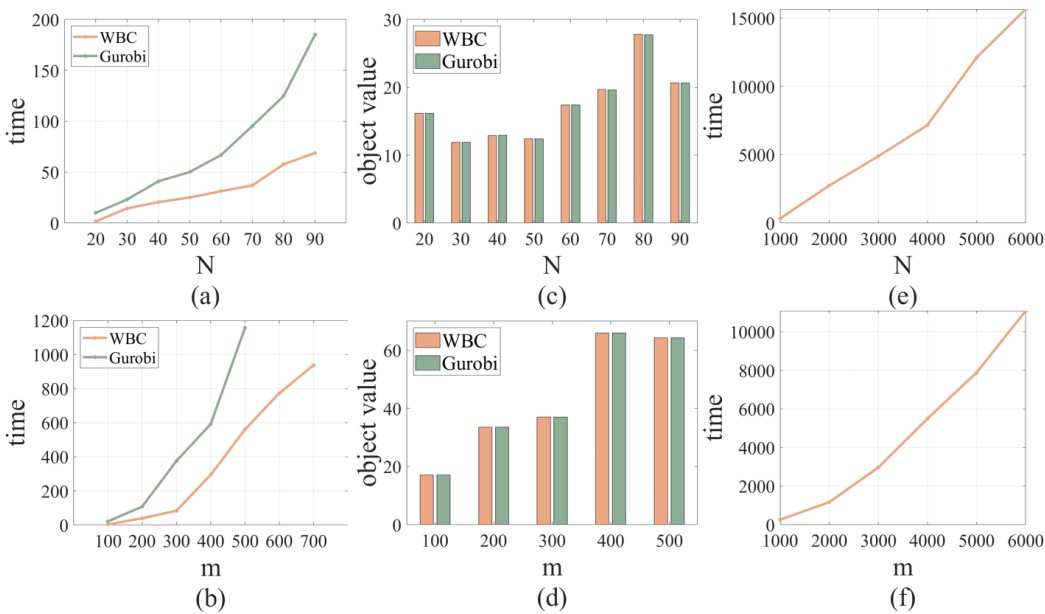

Figure 2: The first two column figures are the computation time and feasibility error of Gurobi and our method. For (a), (c) , $m = 100$. For (b), (d) , $N = 30$. The third column figures are the computation time of our method when the problem scale is very large. For (e) , $m = 50$. For (f) , $N = 10$.

We also illustrate the convergence speed of our algorithm. From Fig. 3 , we can see that our algorithm displays a super-linear convergence rate for the objective value, which is consistent with the result of (Ye et al., 1993).

**Second Experiment** For the performance of fairness, we compare the max Wasserstein distance between WBC and standard WB to input distributions. We divide all distributions into two parts,

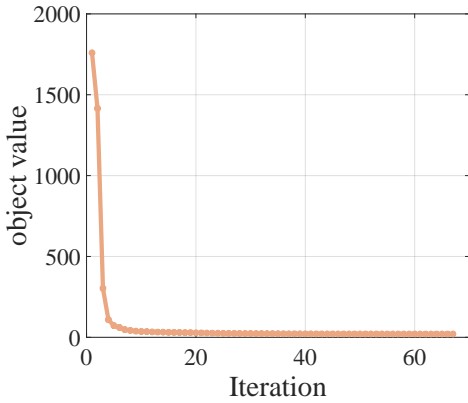

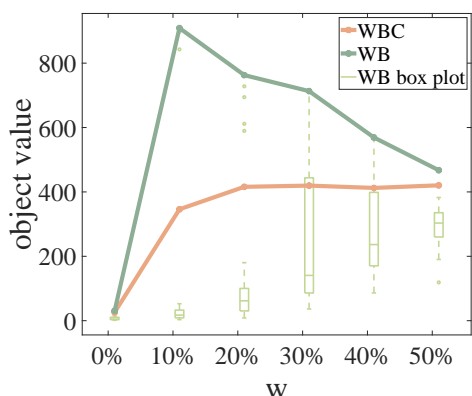

Figure 3: $N = 90, m = 200$. Performance of our algorithm which converges in 67 steps.

Figure 4: $m = 200, N = 30$. Performances of our algorithm and standard WB when distributions are imbalanced.

each part is similar internally, yet very different from the other. Indeed, the measure of the first part concentrates in the first 10 points (among all 200 points), while measure of the second part concentrates in the last 10 points. We demonstrate the fairness performance of these two methods in Fig. 4. An "imbalanced factor" is defined to measure the imbalance between two parts, which represents the proportion of the first part, denoted by $w$. Let "object value" denote the maximum Wasserstein distance between the barycenter and input distributions.

In Figure 4 , we use box plot to represent the distribution of the Wasserstein distance for barycenter to all distributions. We can observe that, when $w = 0\%$, which means all the distributions are similar, or $w = 50\%$, which means the two parts of distributions have same quantity, the cost of standard WB are relatively balanced. When $w = 10\% \sim 40\%$, standard WB has many outliers and an extremely uneven Wasserstein distance distribution. Especially when $w = 10\%$, which means the distributions are extremely imbalanced, some objective values significantly higher than the mean, reaching nearly 900, while most objective values are under 100. Our algorithm effectively eliminates this bias by calculating a barycenter with minimize the maximum Wasserstein distance for individual distributions. We observe a very small difference in the Wasserstein distance over the distributions in our algorithm no matter the distributions are balanced or not. Thus, the object values of our method are always much lower than standard WB.

**Experiments on FairFace Dataset:** For WBC, the objective value denotes the maximum of all Wasserstein distance between WBC and given distributions. For standard WB, the objective value denotes the mean of Wasserstein distance between barycenter and distributions of each races. From Figure 5 , we can observe that the standard WB has a significant gap between the object values in different races, with significantly higher for Middle Eastern. The object value of "Middle Eastern" is 78.30, which far greater than the object value of our algorithm (45.37). At the same time, our algorithm controls the object value within a range only slightly above the mean of WB (38.70).

## 4.2 FAIR ENSEMBLE

*Learning from noisy labels* is one of the fundamental problems in deep learning (Natarajan et al., 2013; Karimi et al., 2020; Song et al., 2022; Karim et al., 2022; Yang et al., 2024), where previous studies use distillation (Kontonis et al., 2024), regularization techniques (Liu et al., 2020; Cheng et al., 2022), teacher model (Han et al., 2018), etc. Those studies has two features: 1). The goal is always trying to select or to create a clean subset of training data; 2) They treat models that only have one data source. What we consider here is to ensemble models trained with different types of noise into one model, such that it gives better predictions than each one of them.

In this experiment, we uses Resnet18 (He et al., 2016) to train 10 classifiers on CIFAR-100. For each classifier, only data labeled in 10 classes are clean, others are added noise with noise rate $u$. For each item, the model outputs an probability vector of dimension 100, each coordinate corresponds to the measure on that label. Inspired by Dognin et al. (2019), where they compute the WB of all predictions, we use WBC as the final probability vector. As is shown in

the table 1, WBC obtains an astonishing accuracy when the noise rate $u$ is 100%, and keeps obtaining better accuracy than WB even though the leading gap shrinks as noise rate declines.

| $u(\%)$ | 100 | 98 | 96 | 94 | 92 | 90 | 80 | 50 | 0 (no noise) |
|---------|-----|-----|------|-----|------|------|-----|-----|--------------|
| WBC | 63 | 54 | 52 | 55 | 56 | 61 | 68 | 75 | 75 |
| WB | 3 | 27 | 39 | 44 | 52 | 58.4 | 67 | 75 | 75 |
| AA | 3 | 22 | 36 | 40 | 46 | 56 | 61 | 75 | 74 |
| Max | 8.3 | 14.8 | 28.4 | 64 | 41.5 | 49.1 | 53 | 67 | 70.3 |

Table 1: Ensemble accuracy with label noises. AA denotes arithmetic average, Max denote the maximum accuracy among models.

## 5 CONCLUSION

We give an efficient algorithm to compute the Wasserstein ball center, outperforming Gurubi on both speed and treatable problem scale. WBC shows better fairness than WB, which makes it more suitable for tasks that is sensitive to minorities, such as model ensembling under imbalanced datasets.

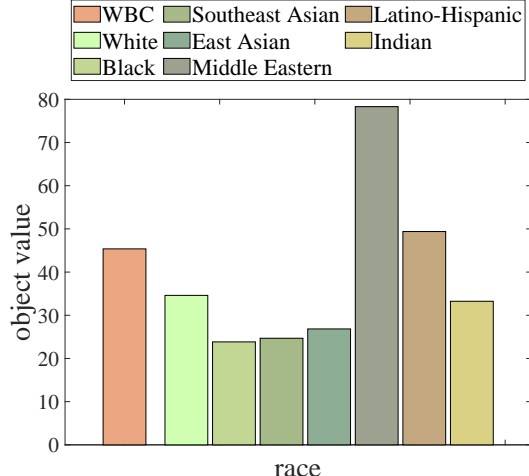

Figure 5: Performance of our algorithm and standard WB on Fairface Dataset. The first column marked "WBC" is the object value of our algorithm, and the others represents the Wasserstein distance from WB to different races respectively.

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

## A    PROOF OF LEMMA 3.1

Lemma 3.1 of (Ge et al., 2019) proved that $A' := \begin{bmatrix} E_1 \\ E_2 \ E_3 \\ \mathbf{1}_m^\top \end{bmatrix}$ has full row-rank, therefore it suffices to prove that the remaining part $A'$ a) has full row-rank. b). $A'\boldsymbol{x} = \boldsymbol{b}$ is equivalent to $\bar{A}' = \bar{b}$.

a). As the $I_N$ in the last $N$ rows og $A$ is of full row-rank, and there are no nonzero terms in the columns of $I_N$, $A$ has full row-rank. b). There is no rows removed from the last $N$ rows. Thus $A\boldsymbol{x} = \boldsymbol{b}$ is equivalent to $\bar{A}\boldsymbol{x} = \boldsymbol{b}$.

## B    ALGORITHM: PREDICTOR-CORRECTOR INNER POINT METHOD

For detailed information, see page. 411 of Wright (1997).

---

**Algorithm 2:** Predictor-Corrector Inner Point Method for Linear Programming

1: **Input**: Linear programming problem in standard form:

$$\min \quad c^T x$$
$$\text{s.t.} \quad Ax = b, \quad x \geq 0$$

where $A \in \mathbb{R}^{m \times n}$, $b \in \mathbb{R}^m$, and $c \in \mathbb{R}^n$.

2: **Initialization**: Set initial feasible point $(x_0, y_0, s_0)$, where $x_0 > 0$, $s_0 > 0$ (dual variables). Choose tolerance $\epsilon > 0$ and set iteration counter $k = 0$.

3:     **while** $\|r_b\| > \epsilon$ *and* $\|r_c\| > \epsilon$ **do**

Compute residuals:

$$r_b = Ax - b \quad \text{(primal residual)}$$
$$r_c = A^T y + s - c \quad \text{(dual residual)}$$
$$r_s = XSe - \mu e \quad \text{(complementarity residual)}$$

where $X = \text{diag}(x)$, $S = \text{diag}(s)$, and $\mu = \frac{x^T s}{n}$ is the duality measure.

4: **Predictor Step**: Solve the linear system for affine scaling direction $(\Delta x^{\text{aff}}, \Delta y^{\text{aff}}, \Delta s^{\text{aff}})$:

$$\begin{bmatrix} 0 & A^T & I \\ A & 0 & 0 \\ S & 0 & X \end{bmatrix} \begin{bmatrix} \Delta x^{\text{aff}} \\ \Delta y^{\text{aff}} \\ \Delta s^{\text{aff}} \end{bmatrix} = - \begin{bmatrix} r_c \\ r_b \\ r_s \end{bmatrix}$$

5: Compute the step size $\alpha_{\text{aff}}$ by finding the maximum step length that maintains $x + \alpha_{\text{aff}}\Delta x^{\text{aff}} \geq 0$ and $s + \alpha_{\text{aff}}\Delta s^{\text{aff}} \geq 0$.

6: **Corrector Step**: Compute the corrector directions using central path perturbation with updated $\mu$:

$$\Delta r_s = XSe - \sigma \mu e$$

and solve the system again to get $(\Delta x^{\text{corr}}, \Delta y^{\text{corr}}, \Delta s^{\text{corr}})$.

7: Compute the total search direction:

$$\Delta x = \Delta x^{\text{aff}} + \Delta x^{\text{corr}}, \quad \Delta y = \Delta y^{\text{aff}} + \Delta y^{\text{corr}}, \quad \Delta s = \Delta s^{\text{aff}} + \Delta s^{\text{corr}}$$

8: Compute the step size $\alpha$ by updating with both predictor and corrector directions.

9: Update variables:

$$x_{k+1} = x_k + \alpha \Delta x, \quad y_{k+1} = y_k + \alpha \Delta y, \quad s_{k+1} = s_k + \alpha \Delta s$$

10: Update the duality measure $\mu$ and increment the iteration counter $k = k + 1$.

11:

12: **Output**: Optimal solution $(x^*, y^*, s^*)$ or termination if stopping criteria met.

---

## C  PROOF OF LEMMA 3.4

Noticing that $A_1 + A_2$ are of a pattern of one simple, easily invertible matrix plus a matrix with low-rank structure, we apply the following

**Lemma C.1.**
$$\bar{W}^{-1} = \bar{W}_1^{-1} - \bar{W}_1^{-1}\mathbf{1}_N(1 + \mathbf{1}_N^\top \bar{W}_1^{-1}\mathbf{1}_N)\mathbf{1}_N^\top \bar{W}^{-1}.$$

*Proof.* This is a corollary of the *Woodbury identity* (Hager, 1989),
$$(P + QLQ^\top)^{-1} = P^{-1} - P^{-1}Q(L^{-1} + Q^\top P^{-1}Q)^{-1}Q^\top P^{-1} \tag{11}$$

for any matrices $P, Q, L$ with legal dimension. $\square$

Now we prove eq. (10) in Lemma 3.4.

*Proof.* Since $Y$ is positive definite, let $Y = U^\top U, U \in \mathbb{R}^{(m-1)\times(m-1)}$. Then $A_2 = (\mathbf{1}_N \mathbf{1}_N^\top) \otimes Y = (\mathbf{1}_N \otimes U^\top)(\mathbf{1}_N^\top \otimes U)$ (Van Loan & Pitsianis, 1993). Thus we have

$$
\begin{aligned}
(A_1 + A_2)^{-1} &= \left(A_1 + (\mathbf{1}_N \otimes U^\top)(\mathbf{1}_N^\top \otimes U)\right)^{-1} \\
&= A_1^{-1} - A_1^{-1}(\mathbf{1}_N \otimes U^\top)(I + (\mathbf{1}_N^\top \otimes U)A_1^{-1}(\mathbf{1}_N \otimes U^\top))^{-1}(\mathbf{1}_N^\top \otimes U)A_1^{-1} \quad (12) \\
&= A_1^{-1} - A_1^{-1}(\mathbf{1}_N \otimes U^\top)(I + \sum_{i=1}^N U A_{ii}^{-1} U^\top)^{-1}(\mathbf{1}_N^\top \otimes U)A_1^{-1} \quad (13) \\
&= A_1^{-1} - A_1^{-1}(\mathbf{1}_N \mathbf{1}_N^\top) \otimes (U^\top(I + \sum_{i=1}^N U A_{ii}^{-1} U^\top)^{-1}U)A_1^{-1} \quad (14) \\
&= A_1^{-1} - A_1^{-1} \left((\mathbf{1}_N \mathbf{1}_N^\top) \otimes (Y^{-1} + \sum_{i=1}^N A_{ii}^{-1})^{-1}\right) A_1^{-1} \quad (15)
\end{aligned}
$$

Eq. (12) comes from Woodbary inequality (11). Eq. (13) and (14) are done by block-wise calculation, since both $A_1$ and $\mathbf{1}_N^\top \otimes U$ are naturally divided into $N$ matrices in $\mathbb{R}^{(m-1)\times(m-1)}$. $\square$

