# OpenReview forum: "An Efficient Algorithm For Computing Optimal Wasserstein Ball Center"
_ICLR.cc/2025/Conference — ICLR 2025 Conference Withdrawn Submission_

### Official Review · Reviewer_qqpB · 2024-10-24

**Soundness:** 2
**Presentation:** 1
**Contribution:** 2
**Rating:** 3
**Confidence:** 4

**Summary:**

Motivated by fairness applications, this paper proposed an alternative notion of the Wasserstein barycenter (WB). The new "center" for probability measures is called the Wasserstein Ball Center (WBC), which aims to minimize the farthest distance from the center to the input measures. The WBC is solved by a linear programming (LP) formulation via an accelerated interior point method tailored to the special low-rank structure of the constraint matrix. Some numeric advantages are demonstrated through an off-the-shelf commercial LP solver Gurobi.

**Strengths:**

Wasserstein barycenter is a timely and important research objective. Minimal distance from the farthest input measure formulation seems to be a new formulation to account for fairness. The proposed interior point method (IPM) uses an observation that the constraint matrix can be expressed as the sum of a block diagonal matrix and a low-rank matrix, which allows faster matrix inversion. The proposal IPM reduces the time complexity of a vanilla IPM from $O(N^3 m^4)$ to $O(N^2 m^3)$, where $N$ is the number of input measures and $m$ is the discretized support size. This is an acceleration of order $O(N m)$.

**Weaknesses:**

${\bf Experiment.}$ The numerical experiment scales in Section 4 are not large. For example, Fig. 2 shows up to $N=6000$ input measures supported on small size $m=50$, and $N=10$ input measures with each support up to $m=6000$ points. Either setup has limited applicability (e.g., low-resolution images, or a very small number of input point clouds). Due to the scaling $O(N^2 m^3)$, I see no reason why the algorithm can be scaled up for a reasonable experimental setup.

${\bf Convergence.}$ The paper claims super-linear convergence for the WBC objective function value (page 8, lines 427-429). However, I cannot see why the super-linear convergence rate in the objective value plot Fig. 3. A log-scale plot of objective value v.s. iteration number should be shown. Claim quadratic convergence consistent with (Ye, Guler, Tapia, Zhang, 1993) is unclear. A quadratic rate should be proved rigorous under certain assumptions in the WBC problem.

Writings of the paper seem to be rushed and many typos are present. Some examples:

-- page 3, line 108: m seems to be the support of WBC and m_i support of the i-th distribution.

-- page 3, line 117: Woodbury's equality (?) [reference link broken].

-- page 5, line 249: primal-duel -> primal-dual

-- page 5, line 258: duel -> dual

**Questions:**

I don't understand the equivalence between the WBC (2) and linear programming (LP) formulation (4). WBC solves an optimization problem $\min_{w, \Omega} \max_{t} \min_{\Pi^{(t)}} [...]$, where $\Pi^{(t)}$ is the coupling between the $t$-th input measure and the Wasserstein center. However, the LP solves a problem $\min_{w, \Omega} \min_{\Pi^{(t)}} \max_{t} [...]$. How was this exchange of the two inner optimization sub-problems achieved?

---

> ### Author Response · Authors · 2024-11-27
>
> Thank you for your comments. Below we address your concerns on our paper.
> > Weakness on applicability
>
> We partially agree with your point on the limitations on applicability. However,  our algorithm has already demonstrated substantial time reduce for users with sufficient computational resources. It is worth mentioning that, to the best of our knowledge, this is the first work to study acceleration algorithms for the WBC problem. As an exploratory effort, we aim to investigate even faster algorithms in future research.
>
> > Convergence
>
> We apply predictor-corrector IPM, whose convergence rate is proved by (Ye, 1993) to be quadratic as long as the feasible region of LP is not empty (See the penultimate paragraph of the introduction of (Ye,1993)). Indeed, the constraint guarentees a feasible solution: for any probability vector $w$, let $\Pi^{(t)}=w\otimes a^{(t)}$, then let $\gamma$ be sufficiently large.
>
> > Question on our LP formulation (6)
>
> Your observation is completely legitimate. Let's break it down step by step. Recall the original objective: $$\min_{w,\Omega}\max_t\min_{\Pi^{(t)}}\langle\Pi^{(t)},D^t\rangle=\min_{w,\Omega}\min_{\Pi^{(t)}}\max_t \langle\Pi^{(t)},D^t\rangle$$
>
>
> For fixed $w,\Omega$, you can see that the choose of $\Pi^{(i)}$ is independent from $\Pi^{(j)}, j\neq i$. In other words, you can minimize every inner product independently, then choose the maximal one among them. So it is safe to change the order of $\Pi^{(t)}$ and $t$. For a even more rigorous proof, we know from the minmax inequality that $\max_x \min_y f(x.y)\leq \min_y \max_x f(x,y)$ always holds, which proves the equality from one side. According to the independence of $\Pi^{(t)}$ mentioned above,  the $\min_{\Pi^{(t)}}\max_{t}$ achieves its value when all $\Pi^{(t)}$ is as small as possible, which gives $\min_{\Pi^{(t)}}\max_t\langle\Pi^{(t)},D^t\rangle\leq \max_t\min_{\Pi^{(t)}}\langle\Pi^{(t)},D^t\rangle$.
>
> We sincerely apologize for writing mistakes you pointed out. All of them are fixed now.
>
> Thank you again for your enlightening questions.

---

### Official Review · Reviewer_NxVe · 2024-10-26

**Soundness:** 3
**Presentation:** 1
**Contribution:** 2
**Rating:** 3
**Confidence:** 4

**Summary:**

The weighted Wasserstein barycenter problem seeks to find a probability distribution $\mu^*$ that minimizes the sum of Wasserstein distances to a given set of probability distributions $\mu_1,\ldots,\mu_N$ with respective weights $w_1,\ldots,w_N$, i.e., $\mu^* = \arg \min_{\mu} \sum_{i=1}^N w_k W_p(\mu, \mu_k)^p$, where $W_p(\cdot, \cdot)$ denotes the $p-Wasserstein distance. The Wasserstein barycenter problem is often used in applications such as image processing, natural language processing, machine learning, and computer graphics.

This paper notes that this formulation of the problem may lead to unfair outcomes towards specific distributions that could represent "protected" subpopulations and thus proposes an alternative formulation to minimize the objective $\min_{\mu} \max_{k=1 \in [N]} w_i W_p(\mu, \mu_k)$. They show that the problem can be formulated as a linear program when the support of the probability distributions is discrete and then describe interior point methods to improve the efficiency of solving the linear program. Finally, the paper includes a number of experiments on synthetic and real-world datasets, evaluating the algorithmic performance for model ensembling on imbalanced data distributions.

**Strengths:**

+ The main technical claims of the paper are supported with mathematically rigorous statements
+ Both the Wasserstein barycenter problem and the notion of fairness are important to the ML community
+ The experiments act as a small-scale demonstration that reinforces the theoretical guarantees provided in the paper

**Weaknesses:**

- The variant of WBC introduced in this paper seems to be the standard socially fair objective, which has been extensively studied for the closely related problem of clustering, often using linear programming techniques, e.g., see [MV21] below
- The discussion on fairness is lacking, given that the main focus of the paper is a problem motivated by fairness
- The main techniques of the paper are for improving interior point methods on the linear program corresponding to the problem, which does not necessarily introduce insightful combinatorial properties about the problem
- The experiments on real-world datasets are not sufficiently comprehensive to convincingly demonstrate a large difference in WBC and the proposed variant (though the motivation is clear) or the practicality of the algorithm across all use cases
- There is a small number of presentation issues, e.g., broken references, extraneous equation markers, missing line breaks, missing theorem statements, substandard proof presentations, etc.

[MV21] Yury Makarychev, Ali Vakilian: Approximation Algorithms for Socially Fair Clustering. COLT 2021: 3246-3264

**Questions:**

N/A

---

> ### Author Response · Authors · 2024-11-27
>
> Thank you for your comments. Your suggestion enhances the completeness of our paper. Below, we address answer to your questions.
>
> > W1 Connections with socially fair clustering
>
> There are similarities in spirit between the WBC and socially fair clustering, since both of them uses $\max_{i\in[N]}$ to replace $\sum_{i=1}^N$ to improve fairness on minority groups. Indeed, part of our idea for the WBC was inspired by socially fair clustering. However, there are fundamental differences between finding a representative on measure space equiped with Wasserstein distance and finding a group of centers on $\mathbb{R}^d$. You distinguish them from all the couplings in the constraints of the LP form of the WBC.
>
> We admit that more discussion and references on socially fair clustering should be involved, so we add them in the revised paper.
>
> >W3 On combinatorial properties of the problem
>
> Your critique is spot-on. We admit that researches on the combinatorial properties would be interesting and valuable. However, the focus of this paper is to design an exact algorithm based on IPM to solve the WBC, where the improvements we made are all based on the properties of the constraint of the WBC problem exclusively. We believe that  the proposed properties in our paper on the constraint matrix will be enlightening for future studies of the combinatorial properties of the WBC.
>
> >W4 On practicality
>
> We partially agree with your point on the limitations of practicality. However, our algorithm has already demonstrated substantial time reduce for users with sufficient computational resources. It is worth mentioning that, to the best of our knowledge, this is the first work to study acceleration algorithms for the WBC problem. As an exploratory effort, we aim to investigate even faster algorithms in future research.
>
> > W5 On presentation issues
>
> We sincerely apologize for writing mistakes you pointed out. All of them are fixed now.

---

### Official Review · Reviewer_12tV · 2024-11-02

**Soundness:** 2
**Presentation:** 3
**Contribution:** 2
**Rating:** 3
**Confidence:** 4

**Summary:**

The paper proposes a method for computing the centre of the _Wasserstein Ball_, that is, the minimum-radius ball in Wasserstein space that contains a given set of distributions. The aim of this Wasserstein Ball Center (WBC) is to be used as an alternative to the Wasserstein Barycentre, as it is expected that WBC will be more robust to outliers and thus be promising in fairness applications.

Though some authors have considered the WBC in the past, the main contribution of the article seems to be the formulation of the min-max problem and the proposed solution.

**Strengths:**

The paper is well motivated. The WBC is certainly an attractive alternative to the Wasserstein barycentre.

The formulation of the optimisation problem is provided in a detailed manner, and it seems to be the main contribution of the article (Thm 3.2). However, this reviewer is not an expert in optimisation and this cannot assess the correctness (other than the experimental evidence provided) or the novelty of the proposed solution

The experimental validation *for the computational complexity of the proposed method* is convincing

**Weaknesses:**

My main reservation with this article is that it makes a number of claims about the proposed method being suitable for *fairness*, however, the method is only either tested synthetically on data with outliers.

Fairness in ML is much more than robustness to outliers. There are defined quantitative indicators for fairness (e.g., disparity impact) and the notion of sensitive/private variables in a learning setting. Modifying an average is *far* from a subset of distributions is not solving a fairness problem. Therefore, saying that *WBC shows better fairness than WB* is a jumping conclusion as i) no fairness problems have been presented in the paper, and ii) no fairness indicators have been measured.

Why not leave the contribution just as a gain in computational complexity? (again, this is not my area of expertise)

There are some English problems, e.g., _we uses_  or "who >> which" but nothing too important. There are also some double parenthesis in the references and missing references (?)

Caption of Fig 1: what is "t"?

There are other types of barycenters that were not considered, e.g., the weak Wasserstein barycenter, and also some unbalanced OT techniques that can help with the outlier detection problem

**Questions:**

see above

---

> ### Author Response · Authors · 2024-11-27
>
> Thank you for your comments! Below, we address the key concern raised in your review.
> > How does the WBC improve fairness?
>
> The WBC we studied is inspired by the concept of socially fair clustering ([1-2]), which aims to minimize the maximum distances from cluster centers to groups, as opposed to traditional clustering that does not consider group separation. This change in objective promotes more equitable solutions across different groups. Similarly, WBC operates in the same spirit: when the probability measures of most distributions are concentrated in one region, critical information from "outliers" with measure distribution distinct from the majority might otherwise be overlooked. WBC addresses this issue, thereby surpassing standard WB in terms of fairness by considering these outliers.
>
> We sincerely apologize for writing mistakes you pointed out, including the confusing "t" in Figure 1. All of them are fixed now. Thank you again for your thoughtful review and for helping us clarify these points.
>
> References
>
> [1] Mehrdad Ghadiri, Samira Samadi, and Santosh Vempala. Socially fair k-means clustering. In
>  Proceedings of the Conference on Fairness, Accountability, and Transparency, pages 438–448,
>  2021
>
> [2] Yury Makarychev, Ali Vakilian: Approximation Algorithms for Socially Fair Clustering. COLT 2021: 3246-3264

---

### Official Review · Reviewer_3B8h · 2024-11-03

**Soundness:** 3
**Presentation:** 2
**Contribution:** 2
**Rating:** 5
**Confidence:** 4

**Summary:**

The paper proposes a method to solve the Wasserstein Ball Center problem in case of distributions with discrete support by rewriting it as a Linear Programming problem. Finally several experiments are conducted to demonstrate the efficiency of the proposed algorithm  as well as the enforced fairness of a solution.

**Strengths:**

The LP formulation of the WBC problem in the discrete support case as well as the proposed iterative algorithm are interesting.

**Weaknesses:**

$\textbf{Experiments}$:

The overall experiments were supposed to demonstrate two points: (i) the effectiveness of the proposed algorithm in solving the WBC problem, (ii) the advantage of WBC over the classical WB in fairness. Both points were not convincingly justified by the chosen experiments.

$\textbf{i}$: For the first point, it would be preferable to see the comparison to some other existing optimization algorithms to solve the problem (6), for example the IPM without preconditioning. Additionally, could you clarify if the results in Figure 2 are averaged over multiple runs, and if not, could you include error bars or standard deviations to show variability.

$\textbf{ii}$:  For the second point, the fairness argument was not very well justified. The fact that a solution to the minmax probelm (2) (WBC) is more sensitive to outliers than a solution to (1) (WB) was straightforward from the fitness function. Are there any quantitative metrics that can justify this claim?

$\textbf{iii}$:  The conclusion was a bit sloppy, would be valuable to list some suggestions for a future work.  For example, you could propose exploring theoretical guarantees, applications to other domains, or comparisons to additional fairness metrics.

$\textbf{Small typos}$:

- Page 3 : missing reference to Woodbury’s equality
- Page 4: missing reference to "fixed-support WB"

**Questions:**

$\textbf{Extensions}$:
- How does the program 6 changes when we lift the assumption of the fixed support distributions ? Since all the distributions are assumed to be of a discrete support, couldn't we just define the common support as a union and lift the assumption ?

$\textbf{Experiments}$:
- How do you explain the non-monotonic behavior of the accuracy as a function of noise ratio that we observe in the table 1.
- Same question for the results at u = 100%
- What happens in the lower noise regime (i.u for u<50%?)
- Have you experimented with the entries following other distribution than uniform.

Could you provide possible explanations or hypotheses for these observations? Additionally, reporting results for the lower noise regime (u < 50%) would provide a more complete picture of the method's performance across different noise levels.

---

> ### Author Response · Authors · 2024-11-27
>
> Thank you for your careful reading and constructive questions! Below, we address the key concerns raised in your review.
> > W1: Other algorithms for the WBC
>
> To our knowledge, the computation of the WBC was not studied before. As a linear programming (LP) problem, there are many practical approaches to solve it, such as the variations of simplex method or interior point methods. In fact, the commercial solver Gurobi, which is compared with our method in section Experiments, implemented a concurrent optimizer that run different state-of-the-art optimization method simultaneously. According to [Gurobi Manual](https://docs.gurobi.com/projects/optimizer/en/current/features/concurrent.html#secconcurrent), "For LP models, ... we currently devote the first concurrent thread to dual simplex, the second through fourth to a single parallel barrier solve, and the fifth to primal simplex. Additional threads are devoted to the parallel barrier solve. Thus, for example, a concurrent LP solve using four threads would devote one thread to dual simplex and three to parallel barrier." **In other words, our proposed algorithm runs faster than all above traditional LP algorithm.**
>
> As for IPM without precondition, there are no reason to quit precondition which reduced the size of problem without any expense. Moreover, IPM relies on precondition to invert some matrix. If there exists this kind of IPM, please inform me.
>
> > W2 Quantative metrics that justify the fairness
>
> A general gap between the maximal distance from WBC to outliers and that of WB can only be 0, for one can construct trivial examples such as identical input distributions. However, this question is both completely legitimate and valuable. We agree that to find a quantitative description for WBC would be helpful for applications. This is a key area we plan to expand on in future work.
> > W3 About conclusion
>
> Very constructive suggestions! We extend the conclusion in the revised version.
>
> >Q1 What happens if we lift the assumption of fixed-support?
>
> When the ground space is infinite such as $\mathbb{R}^d$, the support of WBC do not have to be in the union of common support, if restricted on a fixed support.
>
> >Questions on Experiments
>
> We do not have clue on the non-monotoic behaviour of WBC yet, it may have something to do with the fact that the resnet does learn a fairly vague concept on severely corrupted labels.
>
> When $u<50%$, we think there is no need to compute the accuracy, since all ensemble method under $u=50%$ output close result from that of clean data ($u=0$). That is because the resnet is so powerful that this amount of noise does not affect much. In fact, majority of *learning from noisy label* focuses on *severe label noise* [1].
>
> The common noise for noisy label learning are uniform or asymetric [1]. The asymetric noise means that a label are more likely to be mislabeled into a particular label. For example, "dog" are more likely to be confused with "cat" than other catagories. Since we focuses on ensemble learning from several models, each with their own noise, we did not find a reasonable or intuitive way to design a universally representative noises for asymetric noise. We would like to add discussion on it in future version.
>
> Thank you again for your time and insightful observations.
>
> References
>
> [1] Song, Hwanjun, et al. "Learning from noisy labels with deep neural networks: A survey." IEEE transactions on neural networks and learning systems 34.11 (2022): 8135-8153.

---

### Official Review · Reviewer_eWCn · 2024-11-05

**Soundness:** 4
**Presentation:** 3
**Contribution:** 2
**Rating:** 5
**Confidence:** 4

**Summary:**

This paper proposes the WBC-problem, which is a variant of the well-known Wasserstein-ball problem. Here, given a set of discrete distributions $D_1, D_2, \dots, D_N$  the goal is to compute a distribution (with discrete support over a set of size $m$) that has minimal maximum wasserstein distance from any of these. The main modification from the typical WB-problem is that we seek to minimize the maximum distance, rather than minimize sum weighted sum of the distances. The motivation for doing so is fairness concerns, where averaging a loss may unfairly penalize distributions that correspond to underrepresented or marginalized classes.

This problem can be relatively straightforwardly be framed as a linear program -- in particular, the wasserstein distance between two discrete distributions can be written as the dot product between 2 appropriately chosen matrices, and thus by using a slack variable we obtain a typical linear program. However, the main technical difficulty in this program is the size -- $N$ and $m$ both can be fairly large, and a naive implementation of the interior point method will achieve an $O(N^3m^4)$-running time, which is infeasible.

Thus, the main technical innovation of this paper is a modified interior point method that achieves an $O(N^2m^3)$ running time for this problem. The high level idea is to exploit the shared structure amongst all of the constraint matrices corresponding to the $N$ distributions. Essentailly, interior point methods rely on inverting a matrix, and the matrix being inverted has a block structure that the authors exploit to significantly reduce the running time.

Finally, this work demonstrates that their algorithm achieves better performance on the Fairfaces dataset than naively using the WB-formulation would.

**Strengths:**

This paper offers a fundamental optimization problem and gives a fairly elegant and non-trivial solution to it. In particular, the gains in their running time enable this algorithm to scale to relatively large sets of distributions.

**Weaknesses:**

I found the claim that concerns about fairness motivate this problem to be somewhat tenuous. First, while the relevance of fairness for tasks such as the fairface dataset makes sense, it is unclear to me that for downstream tasks over more complex distributions that the WBC method will significantly improve fairness. In particular, arguing for a lower maximum distance does not ensure fairness in any of the more rigorous definitions of fairness.

Second, while it is intuitive to me that the WBC method offers a reasonable solution in ensuring no class distribution is too far from the mean, it is unclear to me why simply changing the distribution weights over WB is insufficient. In particular, if it appears as though one distribution is significantly underrepresented, we could simply apply a larger weight to that distribution and re-run the original optimization problem. While this is more computationally expensive, it could nevertheless be better suited for downstream tasks where the freedom to choose the weights of each distribution allows the practitioner to achieve some kind of objective measure of fairness.

**Questions:**

See weaknesses -- could you address my concerns about the connection between this problem and fairness?

---

> ### Author Response · Authors · 2024-11-26
>
> Thank you for your comments! Below, we address the key concerns raised in your review.
> > W1: How does the WBC improve fairness?
>
> The WBC we studied is inspired by the concept of socially fair clustering ([1-2]), which aims to minimize the maximum distances from cluster centers to groups, as opposed to traditional clustering that does not consider group separation. This change in objective promotes more equitable solutions across different groups. Similarly, WBC operates in the same spirit: when the probability measures of most distributions are concentrated in one region, critical information from "outliers" with measure distribution distinct from the majority might otherwise be overlooked. WBC addresses this issue, thereby surpassing standard WB in terms of fairness by considering these outliers.
> > W2: Why not simply change the weight of distributions in standard WB?
>
> To determine which distribution is relatively far away from others can be challenging, let alone make assumption on the weights of all distributions. The WBC can avoid those extra computation as well as potential disputes on how to determine those weights by directly minimize the maximum Wasserstein distance to every input distribution.
>
> Thank you again for your thoughtful review and for helping us clarify these points.
>
> References
>
> [1] Mehrdad Ghadiri, Samira Samadi, and Santosh Vempala. Socially fair k-means clustering. In
>  Proceedings of the Conference on Fairness, Accountability, and Transparency, pages 438–448,
>  2021
>
> [2] Yury Makarychev, Ali Vakilian: Approximation Algorithms for Socially Fair Clustering. COLT 2021: 3246-3264

---

### Note · Authors · 2024-11-27

**Comment:**

We would like to express our sincere gratitude to the conference organizers and reviewers for their time and effort in considering our submission.” We truly appreciate the opportunity to present our work to such a prestigious venue.

After the fruitful rebuttal process with the reviewers, we realise that the current version of our paper needs to be perished. While we firmly believe the WBC problem itself is a valuable and promising subject, we have decided to withdraw our submission at this time to allow for further improvement.
We deeply value the support and understanding of the conference committee and look forward to contributing to this esteemed conference in the future. We wish the event great success.

Best regards,

**Withdrawal Confirmation:**

I have read and agree with the venue's withdrawal policy on behalf of myself and my co-authors.